# High Disinfectant Tolerance in *Pseudomonas* spp. Biofilm Aids the Survival of *Listeria monocytogenes*

**DOI:** 10.3390/microorganisms11061414

**Published:** 2023-05-27

**Authors:** Gunn Merethe Bjørge Thomassen, Thorben Reiche, Martinus Hjørungnes, Lisbeth Mehli

**Affiliations:** Department of Biotechnology and Food Science, Norwegian University of Science and Technology (NTNU), 7491 Trondheim, Norway

**Keywords:** food processing environment, antimicrobial tolerance, *Pseudomonas* spp. PAA, multi-species biofilm, *Listeria monocytogenes*

## Abstract

*Pseudomonas* spp. are the most commonly found bacteria in food-processing environments due to properties such as a high growth rate at low temperatures, a high tolerance of antimicrobial agents, and biofilm formation. In this study, a set of *Pseudomonas* isolates originating from cleaned and disinfected surfaces in a salmon processing facility were screened for biofilm formation at 12 °C. A high variation in biofilm formation between the isolates was observed. Selected isolates, in both planktonic and biofilm states, were tested for resistance/tolerance to a commonly used disinfectant (peracetic acid-based) and antibiotic florfenicol. Most isolates showed a much higher tolerance in the biofilm state than in the planktonic state. In a multi-species biofilm experiment with five *Pseudomonas* strains with and without a *Listeria monocytogenes* strain, the *Pseudomonas* biofilm appeared to aid the survival of *L. monocytogenes* cells after disinfection, underscoring the importance of controlling the bacterial load in food-processing environments.

## 1. Introduction

Bacteria within the genus *Pseudomonas* are Gram-negative, rod-shaped, and non-spore-forming obligate aerobes [1]. Members of this highly diverse genus, which has more than 200 described species, have been found in a wide range of environments [2]. They are non-fastidious, have broad temperature tolerance for growth [3], are highly tolerant of various antimicrobial agents [4,5,6,7], and innately produce biofilms [8,9]. Due to their ubiquitous nature, they have frequently been detected in food-processing environments [10,11]. In our recent study [12], we found them to be the most abundant genus in a salmon-processing facility after cleaning and disinfection.

The genus *Pseudomonas* contains many psychrotrophic and psychrophilic species [13], and with a high growth rate at low temperatures, they are likely to outcompete other bacteria. Therefore, even the initially low numbers of *Pseudomonas* spp. in refrigerated food products could cause food spoilage during cold storage due to their rapid growth, and *Pseudomonas* spp. have been suggested to be the predominant spoilage organisms found in aerobically stored and chilled fish products [14,15]. Many *Pseudomonas* strains have been shown to exhibit a high tolerance of disinfectants and are therefore often predominant in food-processing environments after cleaning and disinfection [7,11,16,17,18,19].

The use of disinfectants comprising peracetic acid (PAA) and hydrogen peroxide (H_2_O_2_) has increased in recent years because they have been shown to be highly efficient, broad-spectrum, and environmentally friendly [20,21,22], as after use, PAA can be decomposed into harmless components relatively quickly [23]. As the disinfection routines in food-processing facilities generally do not completely sterilize the processing surfaces, low numbers of bacteria remain after disinfection. The disinfection procedure acts as a selective pressure, favoring the most tolerant cells, which, in turn, proliferate and can potentially contaminate the food product [7].

Many *Pseudomonas* spp. have been shown to carry multidrug-efflux-pump systems, which provide them with a high tolerance of a wide range of antimicrobial agents [5,24,25,26]. A positive correlation between the tolerance of different antimicrobial agents, both disinfectants and antibiotics, has been demonstrated [27,28,29]. This indicates that the survival of the bacteria exposed to sub-lethal concentrations of disinfectants could co-select for both disinfectant- and antibiotic-resistant properties. *Pseudomonas* spp. from food-processing environments have been found to express high levels of resistance to different antibiotics [5,26,30]. The antibiotics of special interest in the salmon-processing environment are those that are commonly used in aquaculture, such as florfenicol and oxolinic acid [31]. Resistance against florfenicol is found to be relatively high (18.5–26.4%) in isolates collected from the sediments at fish farm sites in Chile [32]. The lower usage of antibiotics in Norwegian aquaculture as compared to that in Chile can lead to the assumption that the resistance level in Norway is low [31,33]. However, to our knowledge, no studies have been conducted that describe florfenicol resistance in bacteria associated with farmed-salmon environments in Norway, except our recent study of antibiotic resistance in *Pseudomonas* spp. by disk-diffusion assay [30].

Strains of *Pseudomonas* spp. have been shown to be efficient producers of biofilm, including several protective extracellular matrix components [9,34]. The bacteria living within the biofilm could be exposed to high concentrations of antimicrobials for prolonged periods of time, but they survived due to the sheltering effect of the biofilm matrix. This may have provided favorable conditions for the development of resistance and increased tolerance [35].

Several studies indicate that interspecies interactions in a biofilm could serve as an accelerator for horizontal gene transfer as well as facilitate adaptation to environmental conditions and the subsequent decreased susceptibility to antimicrobials [34,35,36]. Additionally, it has been indicated that multi-species, *Pseudomonas*-dominated biofilms could host and shelter pathogens, such as *Listeria monocytogenes*. Fagerlund et al. [37] showed that *L. monocytogenes*, when paired with *P. putida* and *P. fluorescens*, survived cleaning and disinfection (C&D) routines together better than with other representative isolates in a meat-processing environment. Therefore, the *Pseudomonas* genus has been of special interest as colonizers in food-processing environments, as a food-spoilage organism, and as protectors of food-borne pathogens.

The aims of this study were to (i) investigate the biofilm-forming capability of *Pseudomonas* isolates at a low temperature, relevant to the salmon-processing industry; (ii) study their tolerance, in both planktonic and biofilm states, of relevant disinfectants and antibiotics; and (iii) examine if a multi-species biofilm of several indigenous *Pseudomonas* strains could aid in the survival of the pathogenic bacteria *L. monocytogenes*.

## 2. Materials and Methods

### 2.1. Sampling, Isolation, and Identification of Isolates

All isolates analyzed in this study were collected in a salmon-processing facility, on the same occasions as described in our previous study [12] (Appendix A). Samples were collected early in the morning, before the start of production, on cleaned and disinfected food-contact surfaces. The detection of *Pseudomonas* spp. was performed by spread-plating on *Pseudomonas* CFC selective agar (PA) (CM0559 and SR0103, Oxoid Ltd., Basingstoke, UK) with a following incubation at 25 °C for 48 h. Colonies growing on this agar were considered to be presumptive *Pseudomonas* sp. Pure cultures were made by picking colonies and re-propagating them at least twice before they were frozen in tryptic soy broth (TSB) with 20% glycerol at −80 °C. Identification of the isolates was carried out by sequencing the *rpoD* housekeeping gene, or 16S rRNA gene, as described in a previous study [30].

The isolate of *Listeria monocytogenes* used in the multi-species biofilm experiment was detected in the same salmon-processing plant during routine controls. This isolate was selected for the experiment as it was found to be of sequence type (ST) 8 [38]. This sequence type has been known to persist in food-processing environments and has been associated with food-borne outbreaks of listeriosis from salmon products [39,40].

### 2.2. Biofilm-Forming Capability Testing by Microtiter Biofilm Assay

A total of 36 isolates from our collection did not grow sufficiently at 12 °C and were omitted from the screening. Therefore, a set of 186 isolates were screened for their biofilm-forming capabilities at 12 °C by an in vitro microtiter biofilm assay using 96-well microtiter plates with peg lids (Nunc™ MicroWell™ and Immuno™ TSP Lids, Thermo Scientific) (illustrated in Appendix A). A standardized inoculum of each isolate was prepared by adding colony material to glass tubes pre-filled with 0.9% NaCl to obtain a cell density with the same transmittance as McFarland Standard 1.0 [41]. From the standardized cell suspension, 0.5 mL were inoculated in 14.5 mL ½ TSB (tryptic soy broth) (15 g TSB + 2.5 g NaCl per 1 L broth). To verify the actual cell concentration in each inoculum, serial dilutions were prepared and plated on plate count agar (PCA) plates in triplicate by the microspot technique [42]. Each inoculum was transferred to four wells in a 96-well plate with a peg lid and incubated at 12 °C for 48 h with gentle shaking at 70 rpm.

The optical density at 650 nm (OD_650_) of the planktonic growth was measured before and after incubation. To measure the biofilm formation, the peg lid was transferred to a new 96-well plate containing recovery medium (½ TSB with 1% Tween 20 (0777-1L, VWR Chemicals, Radnor, PA, USA)) and the biofilm was detached from the pegs by sonication (Branson 5800 Ultrasonic Cleaner, Branson Ultrasonics Co., Brookfield, CT, USA) at 40 kHz for 15 min. The OD_650_ in the wells was measured after sonication.

Biofilm formation for each isolate was calculated based on OD measurements as follows: The level of detection (LoD) was defined as mean_control_ + 3 × SD_control_ and was calculated separately for each plate. Isolates with a mean OD_650_ (planktonic) ≤ LoD were characterized as having too poor growth under the given conditions, and their biofilm capability could not be evaluated. OD_650_ (planktonic) and OD_650_ (biofilm) for each isolate were calculated by subtracting the LoD of the respective plate from the mean OD_650_ for the isolate. Because there was a large difference in growth among the isolates under the given conditions, their biofilm-forming capacities were calculated as percentages: (mean OD_650_ (biofilm) − LoD)/(mean OD_650_ (planktonic)-LoD) × 100. Based on this calculation, the isolates were characterized as strong, medium, or weak biofilm producers if the biofilms were >10%, 6–10%, or 0–5%, respectively. If the OD_650_ (biofilm) was below LoD, no biofilm was detected.

### 2.3. Antimicrobial Susceptibility Testing in Microtiter Plates

Among the 186 isolates, 11 *Pseudomonas* isolates (LJP040, LJP042, LJP316, LJP321, LJP760, LJP788, LJP841, LJP863, LJP882, LJP895, and LJP906), one Serratia isolate (LJP847), one Aeromonas isolate (LJP900), and one *L. monocytogenes* isolate (MF4624) [43] were selected for further characterization of in vitro susceptibility to two specific antimicrobial agents. One was the disinfecting agent peracetic acid (PAA) in the form of a commercial product, Aqua DES Foam PAA (H661, Aquatiq Chemistry AS, Lillehammer, Norway). The commercial disinfectant contained 5% peracetic acid, 20% hydrogen peroxide, and 10% acetic acid. The other was the antibiotic florfenicol (F1427, Sigma, Saint Louis, MO, USA), which was the most commonly used antibiotic in Norwegian salmon farming [31]. The selection was based on differences in species classification and biofilm-forming capabilities. The selected isolates were tested in the presence of the antimicrobial agents in order to determine the minimum inhibitory concentration (MIC), the minimum bactericidal concentration (MBC), the minimum biofilm eradication concentration (MBEC), and the log_10_-reduction, according to the procedure described by Harrison et al. [44] and the MBEC™ Assay Procedural Manual, version 1.1 (Innovotech, Inc., Edmonton, AB, Canada), with minor adjustments. The procedure is described briefly here and illustrated in Appendix A.

Inoculum preparation and verification were conducted, as described previously, to a cell density equal to McFarland standard 1.0, followed by plating for the verification of viable cells. Inoculums of each isolate were transferred to 3 full columns each in a 96-well plate with a peg lid and incubated at 12 °C with gentle shaking (70 rpm) for 48 h. After incubation, the peg lid with biofilm formation was transferred to a challenge plate that contained a gradient of a single antimicrobial agent. The challenge conditions used were 15 min for Aqua Des Foam PAA and 24 h for florfenicol, both at 12 °C. During this exposure time, the biofilm on the peg lid shed cells into the broth in the challenge plate. The challenge plate was subsequently separated from the peg lid and used to determine the MIC and MBC. The biofilm on the peg lid was detached from the pegs by sonication (Branson 5800 Ultrasonic Cleaner, Branson Ultrasonics Co., Brookfield, CT, USA) at 40 kHz for 15 min and recovered in a new 96-well plate with recovery medium. The plate with the recovered cells from the biofilm was used to determine MBEC and log_10_-reduction.

Tenfold serial dilutions were prepared, and plating for viable cell count was conducted on PCA using the microspot technique. Plates were incubated at 15 °C for 48 h before counting.

### 2.4. Survival of Listeria Monocytogenes in Multi-Species Pseudomonas Biofilm

An investigation of survival in single- and multi-species *Pseudomonas* biofilms after disinfection under industry-relevant conditions of the disinfectant Aqua DES Foam PAA was performed on a selection of 5 *Pseudomonas* isolates. Of these isolates, three belonged to the *P. fluorescens* group (LJP040, 042, 321), one to *P. putida* (760), and one to *P. lundensis* (788). Selection criteria were different species assignments, biofilm-forming properties, and tolerance properties. Additionally, the fate of *L. monocytogenes* in the multi-species biofilm was also investigated. The methods for biofilm growth and disinfection survival were inspired by the biofilm experiment described by Heir et al. [45], with some adjustments to suit our purposes (Figure 1).

Freshly grown colonies of the 6 selected isolates were transferred into separate sterile glass tubes containing 10 mL of 0.9% NaCl and standardized to the McFarland equivalence turbidity standard 3.0 [41]. A mixed culture of the five *Pseudomonas* isolates was prepared using an equal volume of each standardized suspension. A cell suspension of *L. monocytogenes* was also prepared and standardized to McFarland 3.0. From these cell suspensions, a set of 3 inoculum solutions were prepared: 400 µL of *Pseudomonas* mixed suspension in 19.6 mL ½ TSB w/1% NaCl; 40 µL of *L. monocytogenes* in 19.96 mL ½ TSB w/1% NaCl; and 400 µL of *Pseudomonas* mixed suspension and 40 µL of *L. monocytogenes* in 19.56 mL ½ TSB w/1% NaCl.

Sterile stainless-steel coupons (25 mm × 5 mm × 1 mm, AISI 316) were aseptically placed in the inoculum solutions, with 4 coupons in each culture (*L. monocytogenes* (Lm), *Pseudomonas* spp. mix (Ps), and the combined solution (PsLm)) in addition to a negative control. The bacteria were given 3 h at room temperature (approx. 23 °C) to attach to the coupons. Each inoculum solution was enumerated immediately after bacteria were added by serial dilutions in 96-well plates and microspots on tryptic soy agar (TSA), brilliance *Listeria* selective agar (BLA) and *Pseudomonas* selective CFC agar (PA). The TSA and PA plates were incubated at 25 °C for 24 h, while the BLA was incubated at 37 °C for 24 h.

Inoculated coupons were transferred with sterile forceps into 5 mL Eppendorf Tubes^®^ (Cat. no.: 0030119401, Eppendorf AG, Hamburg, Germany) pre-filled with 3 mL ½ TSB and incubated with gentle shaking (70 rpm) at 12 °C for 72 h. After 72 h of incubation (day 3), enumeration of planktonic cell growth was performed for all samples, including negative controls, and two coupons from each set were disinfected. Coupons were rinsed carefully in 0.9% NaCl before being submerged in a 1% solution of the disinfectant for 15 min. To test bacterial growth and survival in biofilm, one untreated and one disinfected coupon from each culture were transferred into 0.9% NaCl for rinsing before they were transferred into Eppendorf Tubes^®^ containing 3 mL of recovery medium (½ TSB w/1% Tween 20). Samples were sonicated for 15 min at 40 kHz and 24 °C in an ultrasonic bath.

All coupons were transferred to new Eppendorf Tubes^®^ and provided with fresh growth medium. Samples were incubated for another 72 h at 12 °C with shaking at 70 rpm.

After 144 h of inoculation (day 6), samples from all suspensions, including negative controls, were enumerated as previously described. All remaining coupons were sonicated, and the recovery solutions were enumerated. All disinfected coupons were stored for an additional period (4 days, total of 7 days after disinfection) after enumeration to test for survival under the initial detection limit (<10 CFU/mL). Regrowth was recorded as a positive result, and the number of days after treatment was registered. The whole experiment was performed in triplicate, with fresh culture medium and bacterial solutions each time.

### 2.5. Biofilm Disinfection: Survival and Growth

A simplified version of the multi-species biofilm experiment was conducted in 18 parallels to acquire a more confident basis for the evaluation of the survival of biofilm cells after disinfection. Bacterial solutions were standardized and inoculum solutions prepared, as previously described. The inoculum solutions of each culture (Lm, Ps, and PsLm) were added in the wells of 6-well plates (6 wells with each solution), and one stainless-steel coupon was added in each well before incubation at 12 °C for 72 h with shaking at 70 rpm. After 72 h of incubation, all coupons were disinfected as described previously, provided with new growth media (½ TSB) in new 6-well plates, and incubated at 12 °C with shaking at 70 rpm. Each well was closely examined for any increase in turbidity at least once a day for 7 days after disinfection treatment. Additionally, 72 h after disinfection, viable cell counts were analyzed in all suspensions by the traditional plate spreading technique on agar plates (TSA, BLA, and PA). Selective agar media (PA and BLA) were used to differentiate populations in co-culture. Samples that were negative on the first assessment were tested again 144 h (6 days) after disinfection.

## 3. Results

### 3.1. Biofilm-Forming Capabilities

A total of 186 presumptive *Pseudomonas* isolates collected from a salmon-processing plant were screened for biofilm formation at 12 °C. Of these, only 69% were classified as *Pseudomonas,* while the rest belonged to *Aeromonas*, *Acinetobacter*, *Morganella*, *Serratia*, *Shewanella*, *Stenotrophomonas*, and *Pseudoalteromonas*. Ten isolates showed poor growth in a planktonic state and under the given conditions; therefore, their biofilm-forming capabilities were not evaluated. This included all isolates of *Pseudoalteromonas* spp. Of the remaining 176 isolates, 11% (*n* = 19) were characterized as good biofilm producers, 26% (*n* = 45) as medium biofilm producers, 34% (*n* = 59) as weak biofilm producers, and for the remaining 30% (*n* = 53), no biofilm was detected (Appendix A). Among the *Pseudomonas* isolates, 12% (*n* = 15) were characterized as strong biofilm producers, 29% (*n* = 37) as medium biofilm producers, 27% (*n* = 34) as weak biofilm producers, and for the remaining 30% (*n* = 39), no biofilm was detected.

### 3.2. Antimicrobial Susceptibility in Selected Isolates

All the tested isolates exposed to Aqua Des Foam PAA showed MIC values below 1%, except one isolate (LJP042) that had an MIC value of 2% (Table 1). All isolates had MBC values of 1% or lower, while 4 isolates had an MBEC value of 2%, and an additional 4 isolates had an MBEC value of 1%.

For the antibiotic florfenicol, the MIC values ranged from 19 µg/mL to 300 µg/mL for most isolates. Similarly, isolate LJP042 had the highest MIC value, as it was still growing at the highest concentration tested (2400 µg/mL), while another isolate, LJP040, had an MIC value of 2400 µg/mL. The MBC values ranged between 4.68 µg/mL (*L. monocytogenes* MF4524) and 600 µg/mL. A total of 4 isolates had MBEC values of >2400 µg/mL.

The mean log-kill of the biofilm cells after exposure to Aqua Des Foam PAA showed less than 1.0 log CFU/mL reduction for 9 isolates when exposed to the user concentration (1.0%) of the disinfectant (Figure 2). At a 2% concentration of the disinfectant, 7 isolates showed a reduction of more than 5 log_10_, while 6 isolates and *L. monocytogenes* MF4524 were still not reduced by 5 log_10_ units. At a 4% concentration of the disinfectant, 13 isolates were completely inactivated. Only one isolate (LJP042) survived, but it was reduced by 7 log units as compared to the growth controls.

### 3.3. Survival and Growth Dynamics of L. monocytogenes and Pseudomonas spp. in Multi-Species Biofilm

In the multi-species biofilm experiment, viable cell counts for each single- and multi-species culture were analyzed in planktonic state, in biofilm, and in disinfected biofilm after 72 h (day 3) and 144 h (day 6) (Figure 3).

The viable cell counts after 72 h were 9.1 log, 9.0 log, and 9.2 log CFU/mL in planktonic state and 6.9 log, 5.5 log, and 7.1 log CFU per coupon of biofilm growth for Ps, Lm, and PsLm, respectively. After 144 h of incubation with a renewal of the growth medium on day 3, the viable cell counts in planktonic state were 9.1 log, 8.9 log, and 9.3 log CFU/mL, with biofilm populations of 6.6 log, 6.5 log, and 6.8 log CFU/mL for Ps, Lm, and PsLm, respectively.

In its planktonic state, *L. monocytogenes* grew considerably less in co-culture with *Pseudomonas* spp. (PsLm) compared with a *L. monocytogenes* monoculture. Based on viable cell counts, *L. monocytogenes* accounted for approximately 0.8% of the total population in planktonic PsLm at day 3. On day 6, *L. monocytogenes* in planktonic state had decreased to approximately 0.6% of the total population. In co-culture biofilm, *L. monocytogenes* accounted for 0.9% of the PsLm population on day 3, while on day 6, this had increased to approximately 2.6% of the population.

Viable cell counts in disinfected biofilm were analyzed by plating the recovered cells immediately after sonication. No growth was observed in any of the disinfected coupons, indicating that the number of surviving cells was below the detection limit (30 viable cells/coupon). After the re-incubation of these cell suspensions for three additional days, only the PsLm culture exhibited regrowth in two replications.

### 3.4. Survival in Disinfected Multi-Species Biofilm

Based on the inconsistent results of the multispecies biofilm disinfection experiment, where survival after disinfection was observed for 2 out of the 3 PsLm cultures, a simplified version of the experiment was carried out in 18 parallels. The biofilms of *L. monocytogenes*, *Pseudomonas* spp. Mix, and a combination were grown on stainless-steel coupons for 3 days before disinfection of the coupons, followed by a transfer into fresh growth medium and further incubation. The cultures were visually inspected for new growth by observing visible increases in turbidity once per day. No increased turbidity was observed in any of the cultures one day after disinfection. On the second day, 4 of the 18 Ps cultures and 7 out of the 18 PsLm cultures showed visible growth, while on the third day, 13 of the Ps cultures and 9 of the PsLm cultures showed visible growth. On day 3 after disinfection, all cultures were plated on TSA, BLA, and PA to test for growth. In this viable cell count test, 10 out of the 18 PsLm cultures showed growth of *L. monocytogenes* (Figure 4).

## 4. Discussion

In this study, 186 bacterial isolates from *Pseudomonas* CFC agar were screened for biofilm-forming capability at 12 °C and graded as strong, medium, weak, or not biofilm-producers. This screening revealed that among the *Pseudomonas* isolates, 12% were characterized as strong biofilm producers, 29% as medium biofilm producers, and 27% as weak biofilm producers. These findings corresponded to previous studies where *Pseudomonas* spp. had been found to be good biofilm producers [8,46]. Liu et al. [47] reported that certain strains of *Pseudomonas* produced biofilm faster at low temperatures (4–10 °C) than at 30 °C. In this study, we used an incubation temperature of 12 °C to simulate the conditions in a salmon-processing facility, and this probably induced efficient biofilm formation in many of the *Pseudomonas* isolates. In this experiment, 29% of the *Pseudomonas* isolates did not produce any detectable biofilm. This did not appear to be related to the species, as we observed diverse species assignments among the non-biofilm group, weak-biofilm group, and medium-biofilm group. Furthermore, we observed that 57% of the *Pseudomonas* isolates with no biofilm formation also had poor general growth under the selected conditions, indicating that the possible biofilm might be below the detection limit of this method. The screening of the biofilm-forming capabilities was conducted by measuring the optical density at 650 nm (OD_650_) and classifying the isolates as strong, medium, or weak biofilm producers. We set our own limits based on the OD_650_ (biofilm) relative to OD_650_ (planktonic) and calculated as previously explained. This means that these characterizations are only relative to the isolates in this study. The OD measurement has some limitations that are important to be aware of. Firstly, the size and shape of the cells, in addition to intra-species aggregation and flocculation, will affect the OD measurement. Another limitation is the sensitivity of the OD measurement, as previously reported by Biesta-Peters et al. [48].

Good cleaning and disinfection routines in food-processing facilities are of high importance, and it is essential that the disinfectants are efficient. The goal of disinfection routines is not necessarily to sterilize the surface but, instead, to keep total bacterial contamination to a minimum and eradicate pathogens. When considering the efficiency of a disinfectant, it must provide a minimum of a 5 log_10_-reduction in cell numbers of the specific test organisms, usually performed on bacteria in a planktonic state [49,50].

Among the isolates tested for tolerance to a PAA-based disinfectant and antibiotic used in fish farming, the *Pseudomonas* isolates showed higher tolerance to both the disinfectant and the antibiotic as compared to the bacteria of the other genera. This was especially evident for bacteria in a biofilm state. Interestingly, the variations in the tolerance patterns between isolates were high, even in different isolates of the same species, indicating that antimicrobial tolerance was not necessarily a strain-specific property. This experiment showed that all the tested isolates had MIC and MBC values below the user concentration of 1%, except for a *P. fluorescens* isolate (LJP042), which had an MIC value of 2%. Both the MIC and MBC values were measured on planktonic state bacteria. The MIC values for disinfectants are of less relevance for the food industry because the disinfected surfaces are usually rinsed with water to prevent disinfectant residue in the food products, and the resident bacteria on the surfaces will therefore not be exposed to the disinfectant for a prolonged period. On the contrary, the MBC and MBEC values are of high interest, as these could better reflect the scenario in a food-processing facility with a short exposure time. In our experiment, 4 isolates had an MBEC value of 2.0%, and an additional 4 isolates had an MBEC value of 1.0%. This indicates that they are likely to survive the disinfection procedure applied in a food-processing facility (1.0% disinfectant, exposure time 15 min).

To reach the 5 log_10_-reduction of the bacteria, different combinations of the disinfectant concentration and exposure times could be used. In this experiment, the exposure time of the biofilm cells to the disinfectant was 15 min. At a 1.0% concentration, only 1 isolate was reduced by 5 log_10_ units. At a 2.0% concentration, 7 isolates were sufficiently reduced, while at a 4.0% concentration, all isolates were reduced by a minimum of 5 log_10_ units. However, a single *P. fluorescens* isolate (LJP042) still had some viable cells. The manufacturer’s recommendation for the disinfectant was a concentration of 1.5–3.0% with an exposure time of 5–15 min, while the salmon-processing facility had chosen a strategy of 1.0% for 15–20 min for daily routines, with higher concentrations occasionally. The results from the susceptibility experiment indicated that the highest concentrations recommended by the manufacturer were sufficient to eradicate most bacteria, also in a biofilm state. However, it must be mentioned that the biofilms in this experiment were single-species biofilms that had been allowed to establish over just 24 h. A multi-species biofilm established over a longer time period has other attributes that could lend it more resistance [37,51]. Fagerlund et al. [37] demonstrated that C&D routines could induce a shift in the bacterial composition in complex multi-genus biofilms, while biofilms only exposed to water remained stable in composition, which could have further implications on biofilm survival and pathogen transmission.

The facility’s strategy of using lower concentrations of the disinfectant than recommended must also be addressed. Our results demonstrated that at 1.0% concentration, the reductions in 9 of the isolates were less than 1 log_10_-unit. This strategy not only results in surviving bacteria but also provides a selective pressure that favors the more tolerant individuals. Previous studies indicate that exposing bacteria to sub-lethal concentrations of a disinfectant can cause co-selection of antibiotic-resistant bacteria and also induce increased tolerance to certain antibiotics [28].

Florfenicol is the most frequently used antibiotic in Norwegian aquaculture [31] and is used to treat infections caused by *Aeromonas salmonicida* and *Pseudomonas anguilliseptica* in both salmon and lumpfish, among others [52,53]. The development of resistance in aquaculture-related bacteria is therefore of high interest. In contrast to disinfectants, antibiotics are meant to have a longer exposure time; hence, the longer exposure time (24 h) for florfenicol. Additionally, the MIC values are more relevant than the MBC and MBEC values as the MIC values are measured after a long exposure time. The susceptibility tests with florfenicol showed MIC values throughout the whole spectrum of the concentrations that were tested. The remarkable high tolerance of the isolate LJP042 (MIC >2400 µg/mL) and the isolate LJP040 (MIC = 2400 µg/mL) has, to our knowledge, not been reported earlier. Both Miranda and Rojas [32] as well as Adesoji and Call [54] reported high MIC values (>512 to >1024 µg/mL) for florfenicol among *Pseudomonas* spp., associated with Chilean salmon farms, while Ho et al. [55] reported MIC values between 0.78 and >100 µg/mL. All the *Pseudomonas* isolates included in this study had previously been characterized as resistant to florfenicol by the disk diffusion method in our recent study [30].

The MBEC values for florfenicol in the studied isolates were generally high, with 4 isolates showing an MBEC value > 2400 µg/mL and 2 isolates with an MBEC value of 2400 µg/mL. Generally, our isolates had much higher MBEC values than MIC values, which supported the findings in previous studies where bacteria had drastically increased their tolerance to antibiotics in biofilms as compared to their response in planktonic states [56,57]. This was particularly expressed in the isolate LJP863, which had relatively low MIC and MBC values (75 µg/mL and 19 µg/mL, respectively) but an MBEC value of >2400 µg/mL. In addition, isolate LJP760 had a relatively low MIC value (150 µg/mL). This isolate was characterized as susceptible to florfenicol in a disk diffusion assay [30]. With an MBEC value of >2400 µg/mL, the protective effect of the biofilm was demonstrated. It was, however, observed that the colonies produced by the cultures exposed to a high concentration of florfenicol were smaller after exposure than the colonies produced by cultures exposed to lower concentrations. This indicated that florfenicol caused a growth delay in the cells recovered from the exposed biofilm, which can be explained by the bacteriostatic effect of florfenicol. A similar phenomenon had been observed in *P. fluorescens* isolates exposed to gradually higher concentrations of the disinfectants dodecyl-dimethylammonium chloride (DDAC) and coco-trimethylammonium chloride (CTC) [7].

In the multi-species biofilm experiment, we observed that *L. monocytogenes* grew considerably less dense in co-culture with *Pseudomonas* spp. (PsLm) as compared to monoculture (Lm) in planktonic state. This could be explained by *Pseudomonas* spp. having a higher growth rate than *L. monocytogenes* under the applied conditions. In the PsLm culture, *L. monocytogenes* accounted for approximately 0.8% of the total population on day 3 and 0.6% on day 6. In the PsLm biofilm, *L. monocytogenes* accounted for 0.9% of the PsLm population on day 3, which then increased to 2.6% of the population on day 6. This increase indicated that *L. monocytogenes* had the potential to persist in such biofilms, though in lower numbers than its cohabitants. Fagerlund et al. [37] demonstrated the same tendency after *L. monocytogenes* had been grown in multi-species biofilms on conveyor belt coupons. The *L. monocytogenes* biovolume increased from day 4 to day 7 of their experiment as compared to the biovolumes of the multi-species biofilms. In the same study, by confocal laser scanning microscopy (CLSM), they also documented that *L. monocytogenes* was present as single cells or small clusters dispersed in the multispecies biofilm and that the density of these cells was higher towards the inner layer of the biofilm. A number of possible mechanisms involved in the increased tolerance to disinfectants and how this can aid the survival of, e.g., pathogens have been discussed by Bridier et al. [51]. Based on the increase of *L. monocytogenes* in the biofilm over time and the simultaneous decrease of *L. monocytogenes* in suspension seen in our study, we suspect that *L. monocytogenes* migrated to the inner layers of the biofilm over time and, as a result of this, is more sheltered against the disinfectant.

To assess the efficiency of the disinfectant Aqua DES Foam PAA, the biofilms of the different cultures, Ps, Lm, and PsLm, were exposed to a 1% concentration of the disinfectant for 15 min. The viable cell counts from the disinfected biofilms were analyzed by plating the recovery suspension immediately after disinfection and sonication. No growth was observed in any of the disinfected coupons at this point. However, after re-incubation of these recovery suspensions for 3 days with the steel coupon still present, regrowth occurred in the PsLm culture in one of the replicates. This indicated that either some cells had survived the disinfection but the numbers were below the detection limit of 30 viable cells/coupon or the cells had entered a viable but not culturable (VBNC) state. After plating on selective agar 3 days post-disinfection, we found that both *Pseudomonas* spp. and *L. monocytogenes* had survived the treatment. Similar results, with regrowth only in the mixed culture, were observed in a similar preliminary experiment. To further investigate the inconsistent disinfection survival characteristics, a simplified version of the experiment was conducted with a higher number of parallels. In this experiment, no viable cells could be detected by spread-plating immediately after disinfection. However, substantial regrowth after two, three, and four days could be observed by the visually increased turbidity in several specimens. None of the single-species biofilms of *L. monocytogenes* survived the disinfection treatment, but in total, 15 of the 18 *Pseudomonas* spp. biofilms and 13 of the 18 PsLm biofilms survived the treatment. *L. monocytogenes* survived in 10 of the PsLm biofilms. This supported the hypothesis that the properties of *Pseudomonas* spp. in biofilm formation and tolerance towards disinfectants also aided pathogens such as *L. monocytogenes* in surviving the disinfection treatment.

Despite this strong indication, there were some inconsistencies in the results. A total of 3 out of the 18 Ps cultures and 5 of the 18 PsLm cultures did not survive the disinfection treatment. In addition, we observed a variation in the colony morphology after spread-plating, suggesting that different strains of *Pseudomonas* sp. dominated in different parallel cultures. This indicated that there were additional factors affecting the survival of the biofilm beyond those investigated in this study. These factors could be that the steel coupons had microscopic irregularities, scars, or scratches on their surfaces, enabling the bacterial cells to evade disinfection. As well, this could be likely in real-world food-processing environments. The topography of the biofilm can also vary between cultures, providing differences in the diffusion of the disinfectant. It is likely that a combination of the above-mentioned factors gives small margins for survival in biofilm after disinfection.

This experiment demonstrated that a commonly used disinfection strategy was sufficient for reducing biofilm populations to a level below detection limits. However, we observed a substantial level of regrowth after 48–96 h, indicating that the biofilm populations were not totally eradicated. The study also demonstrated how an often-overlooked bacterial genus such as *Pseudomonas* could indirectly threaten food safety by sheltering low numbers of pathogens in its biofilm matrix.

When performing these experiments, we attempted to provide conditions as close as possible to a real-world scenario. However, there were some crucial differences between the disinfection process in this study and the cleaning and disinfection procedures in real life. In the food-processing facilities, mechanical forces, including swabbing and scrubbing surfaces, and chemical cleaning agents are used. These actions were not conducted in our experiments. However, swabbing and scrubbing may not be applied to all surfaces consistently, and some surfaces and areas could be overlooked due to human error or difficult access. Therefore, the conditions in the experiments were not unrealistic.

## 5. Conclusions

Species of the genus *Pseudomonas* are commonly found in food environments, and some of them are recognized as important food spoilage bacteria. However, in terms of food safety, members of this genus have often been overlooked as they are not directly associated with food-borne infections in humans.

In this study, we demonstrated the variation in biofilm-forming capacity at given conditions in different isolates originating from a salmon processing facility. We also demonstrated the variation in tolerance to the PAA-based disinfectant, Aqua Des Foam PAA, regularly used in food processing facilities, and one antibiotic relevant to aquaculture, revealing that many *Pseudomonas* isolates have a high inherent tolerance to the disinfectant, especially in biofilm state. The resistance to florfenicol was also high in several isolates, with MIC values of 2400 μg/mL and higher. Ultimately, we showed how the biofilm-forming and disinfectant tolerance of *Pseudomonas* spp. can aid the survival of *Listeria monocytogenes*. By doing so, *Pseudomonas* spp. residing in the food processing environment indirectly threatens food safety.

## Figures and Tables

**Figure 1 microorganisms-11-01414-f001:**
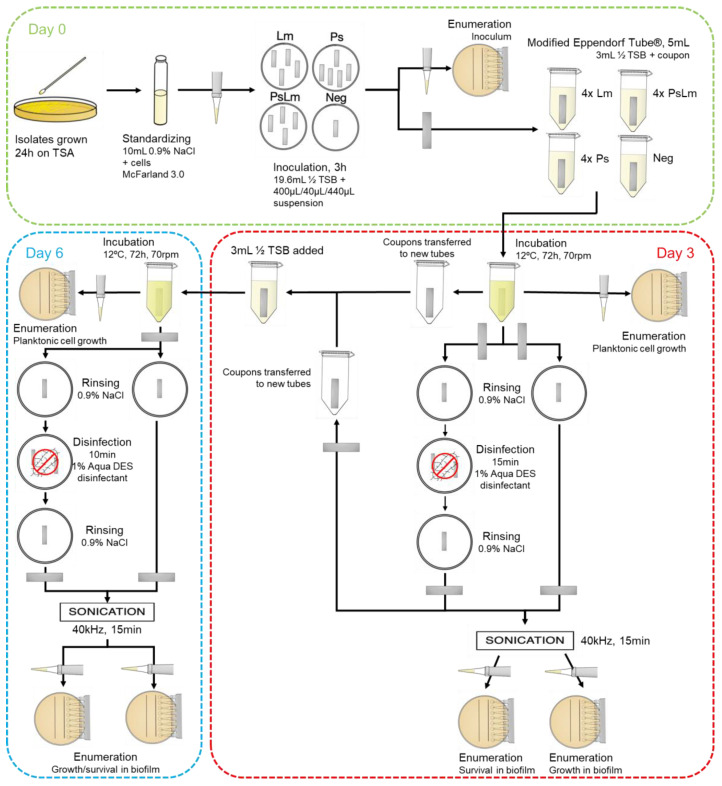
Flowchart of the biofilm disinfection experiment. Freshly grown colonies of *L. monocytogenes* and *Pseudomonas* spp. isolates were suspended in a 0.9% NaCl solution and standardized to McFarland 3.0. Stainless steel coupons were submerged in the cell suspensions for 3 h at room temperature. Inoculum solutions: Lm = *L. monocytogenes*; Ps = *Pseudomonas* spp. Mixture; PsLm = *L. monocytogenes* and *Pseudomonas* spp. co-culture. Inoculated coupons were transferred to Eppendorf Tubes^®^, containing 3 mL ½ TSB, and incubated for 72 h at 12 °C with gentle shaking (70 rpm). After incubation, half of each set was disinfected (1% Aqua DES Foan PAA disinfectant, 15 min). One disinfected and one not disinfected coupon of each set was incubated again (under the same conditions). Enumeration of viable cells was performed by the microspot technique for inoculum, planktonic cells, and detached biofilm at the start, on days 3 and 6.

**Figure 2 microorganisms-11-01414-f002:**
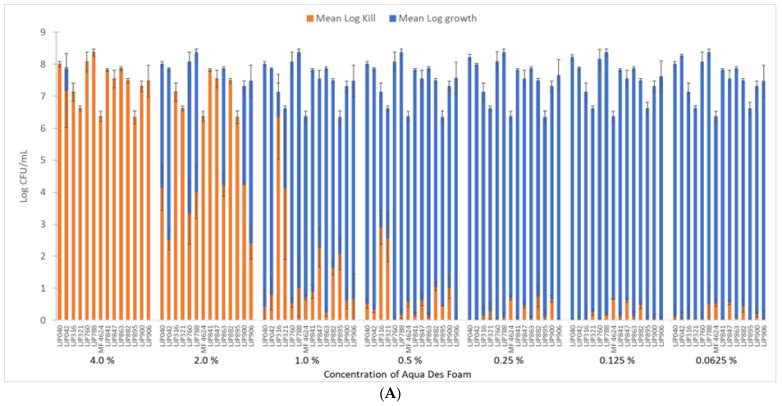
Log_10_-reduction of 14 isolates (LJP040, LJP042, LJP316, LJP321, LJP760, LJP788, MF4624, LJP841, LJP847, LJP863, LJP882, LJP895, LJP900, and LJP906) after exposure to (**A**) the disinfectant Aqua Des Foam PAA at different concentrations for 10 min and (**B**) the antibiotic florfenicol at different concentrations for 24 h.

**Figure 3 microorganisms-11-01414-f003:**
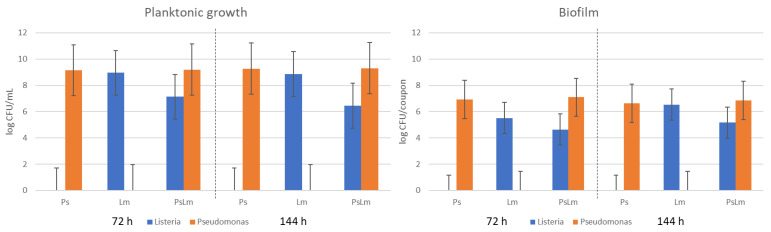
Viable cell counts from multi-species cultures for planktonic growth and biofilm growth. The values are shown as an average between three technical replications. Ps denotes the *Pseudomonas* culture consisting of 5 different strains; Lm denotes the *L. monocytogenes* in a single culture; and PsLm denotes the combined culture of *Pseudomonas* mix and *L. monocytogenes*.

**Figure 4 microorganisms-11-01414-f004:**
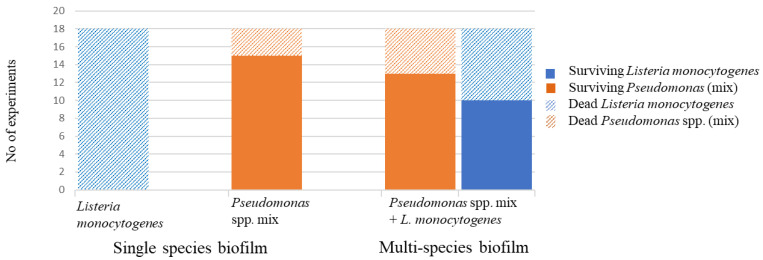
Test for survival on disinfected stainless-steel coupons. Steel coupons inoculated with *L. monocytogenes* (Lm), a mix of 5 *Pseudomonas* spp. (Ps), and a combination (PsLm) were added in 5 mL ½ TSB to the wells of 6-well plates and incubated at 12 °C for 72 h (3 days) before disinfection. The bar graph shows the number of biofilm cultures that still contained viable cells after the disinfection. All 18 cultures of *L. monocytogenes* died after disinfection. Of the 18 *Pseudomonas* spp. cultures, 15 still contained viable cells after disinfection. In the cultures with both *Pseudomonas* spp. and *L. monocytogenes*, 13 cultures contained viable cells of *Pseudomonas* spp., and ten cultures contained viable *L. monocytogenes* cells after disinfection.

**Table 1 microorganisms-11-01414-t001:** Overview of the measured MIC, MBC, and MBEC values for the 14 isolates after exposure to Aqua Des Foam PAA for 15 min at 12 °C and to florfenicol for 24 h at 12 °C. The marking “-” indicates that the OD was below the limit of detection.

Isolate ID	Predicted Species	Biofilm Capability	Aqua Des Foam PAA (%)			Florfenicol (µg/mL)		
			MIC	MBC	MBEC	MIC	MBC	MBEC
LJP042	*P. fluorescens*	Strong	2.0	0.5	2.0	>2400	600	>2400
LJP316	*P. fluorescens*	Medium	0.25	0.25	0.5	300	300	2400
LJP321	*P. fluorescens*	Medium	0.25	0.25	0.5	300	300	2400
LJP760	*P. putida*	Low	0.25	1.0	2.0	150	150	>2400
MF4624	*L. monocytogenes*	-	-	0.06	-	-	4.68	-
LJP040	*P. fluorescens*	No	0.25	0.5	2.0	2400	300	600
LJP788	*P. lundensis*	Low	0.25	1.0	2.0	300	300	>2400
LJP863	*Pseudomonas* sp.	Low	0.06	0.25	1.00	19	75	>2400
LJP906	*P. fluorescens*	No	0.06	-	1.00	75	19	150
LJP882	*P. libanensis*	No	0.06	-	0.50	300	600	600
LJP841	*Pseudomonas* sp.	Medium	0.06	0.25	1.00	300	600	600
LJP895	*P. veronii*	Strong	0.06	-	0.25	19	75	75
LJP847	*S. liquefaciens*	No	0.06	0.13	0.50	300	600	300
LJP900	*A. hydrophila*	Medium	0.13	0.13	1.00	-	19	75

## Data Availability

The data presented in this study are available in this article and the Appendix A.

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
