# Peer review of "High Disinfectant Tolerance in *Pseudomonas* spp. Biofilm Aids the Survival of *Listeria monocytogenes"

_microorganisms, 2023, doi:10.3390/microorganisms11061414_

Round 1

Reviewer 1 Report

In this manuscript, the authors evaluated the biofilm forming ability of Pseudomonas isolates under low temperature conditions, which mimics the condition in food processing facilities. The authors reported MIC, MBC, MBEC, and Killing of PAA and Florfenicol for representative strains. Pseudomonas and Listeria monocytogenes are both common pathogens in the food industry. The authors evaluated the composition of multi-species biofilm and found that even though the number of Listeria was much less in the multi-species biofilm, there was a higher chance for Listeria to survive disinfection compared to the monoculture. This observation is intriguing and opens up future studies on the mechanisms by which Pseudomonas protects Listeria in a biofilm. I have a couple of questions and comments on some specific experiments in this manuscript:

1.     The first paragraph of the introduction and discussion sections should be deleted.

2.     Table 1, what do “-“ marks mean? Were they not tested for some reason or beyond the limit of detection?

3.     Section 3.2, how do the authors explain the discrepancy between Mean Log Kill and MBEC? With 2% PAA, some isolates were not fully killed based on Mean Log Kill. However, Table 1 suggests none of the isolates was able to recover growth resulting in a MBEC lower or equal to 2%.

4.     In the latter half of the work, the authors used a mixed Pseudomonas to study the multispecies biofilm. I have a couple of questions related to this mixed culture:

1)    Were different isolates (different Pseudomonas isolates as well as Listeria) mixed at equal number? And did the total number match that of the monoculture?

2)    How are the biofilm capability and antimicrobial susceptibility of the Pseudomonas mix compared to single isolates?

3)    How were the mixed species biofilms disinfected? Figure 1 indicates 1% PAA but lines 221-222 indicates a mixture of peracetic acid, hydrogen peroxide and acetic acid. If 1% PAA was used here, how do the authors explain that the MBEC of mixed Pseudomonas was lower than single isolates reported in Table 1?

Author Response

Response to reviewers:
The authors thank the reviewers for a thorough review of our paper and for providing constructive and
valuable comments. We value all the feedback given and have done our best to make improvements
and clarifications according to each comment. Below you will find all the comments we received,
together with our answers on how we have addressed each of them.
One reviewer asked for English editing of the manuscript. This has been done by MDPI’s Author
services. Changes has been made throughout the whole manuscript, according to these suggestions,
to improve the language. Changes due to English editing is not marked in the re-submitted
manuscript.
Reviewer #1
In this manuscript, the authors evaluated the biofilm forming ability of Pseudomonas isolates under
low temperature conditions, which mimics the condition in food processing facilities. The authors
reported MIC, MBC, MBEC, and Killing of PAA and Florfenicol for representative strains.
Pseudomonas and Listeria monocytogenes are both common pathogens in the food industry. The
authors evaluated the composition of multi-species biofilm and found that even though the number of
Listeria was much less in the multi-species biofilm, there was a higher chance for Listeria to survive
disinfection compared to the monoculture. This observation is intriguing and opens up future studies
on the mechanisms by which Pseudomonas protects Listeria in a biofilm. I have a couple of questions
and comments on some specific experiments in this manuscript:
1. The first paragraph of the introduction and discussion sections should be deleted.
Paragraphs deleted.
2. Table 1, what do “-“ marks mean? Were they not tested for some reason or beyond the limit
of detection?
The marking “-“ means that the OD was below the limit of detection. This explanation is added
in the table text.
3. Section 3.2, how do the authors explain the discrepancy between Mean Log Kill and MBEC?
With 2% PAA, some isolates were not fully killed based on Mean Log Kill. However, Table 1
suggests none of the isolates was able to recover growth resulting in a MBEC lower or equal
to 2%.
Mean Log Kill and MBEC are measured in the same cell suspension. However, there are
some discrepancies in our results here and, the authors believe there are two reasons for
this. 1) The MBEC result is based on OD measurement which we know has a limitation in the
sensitivity. This is commented in our discussion (lines 384-385). The Mean log kill analysis is
based on serial dilutions and plating og the dilutions on agar plates, and therefore has a
higher sensitivity. 2) The OD measurement in the MBEC analysis was done after 24 h
incubation at 12 C, while the agar plates for the mean log kill analysis were read after 48 h
incubation at 15 C. We acknowledge that we should have adjusted the MBEC procedure at
this point, and incubated for 48 h.
4. In the latter half of the work, the authors used a mixed Pseudomonas to study the multispecies
biofilm. I have a couple of questions related to this mixed culture:
1) Were different isolates (different Pseudomonas isolates as well as Listeria) mixed at equal
number? And did the total number match that of the monoculture?
To make the standardized cell suspensions we measured the turbidity and compared to McFarland
standards. A mixed culture of the five Pseudomonas isolates was prepared using an equal volume of
each standardized suspension. Since Pseudomonas spp. cells (about 1–5 μm long) are larger than
Listeria spp. (0.5–2 μm long) a cell suspension of Listeria spp. equal to McFarland 3.0 will have a
slightly higher cell count than a cell suspension equal to McFarland 3.0 of Pseudomonas spp. To
adjust for this, we used 10-fold lower volume of the Listeria suspension than the Pseudomonas spp.
mix. The Pseudomonas spp. mix was a mixture of equal volumes of the standardized cell
suspensions of each Pseudomonas strain. CFU/mL was verified for each culture at starting point by
serial dilution and plating by microspot technique.
2) How are the biofilm capability and antimicrobial susceptibility of the Pseudomonas mix compared
to single isolates?
We did not measure the biofilm capability of the Pseudomonas mix the same way as we did for the
single isolates. Seen in retrospect, it would have been a reasonable measurement to perform.
3) How were the mixed species biofilms disinfected? Figure 1 indicates 1% PAA but lines 221-222
indicates a mixture of peracetic acid, hydrogen peroxide and acetic acid. If 1% PAA was used here,
how do the authors explain that the MBEC of mixed Pseudomonas was lower than single isolates
reported in Table 1?
The authors accept that this was not well explain and have now made changes to clarify (line 150-
151). In this study a commercial product (Aqua DES Foam PAA), with PAA as main active ingredient,
was used. This commercial product contains 5% PAA, 20% hydrogen peroxide and 10% acetic acid.
When a 1% solution of the commercial product was used, this contained 0.05% peracetic acid, 0.2%
hydrogen peroxide and 0.1% acetic acid. This explanation is added in chapter 2.3 (the first-time use
of Aqua DES Foam PAA is mentioned). In Figure 1 we have changed it to 1% Aqua DES.

Reviewer 2 Report

In the presented study the authors collected Pseudomonas species on cleaned and disinfected food contact surfaces and investigated the strains for resistance to commonly used disinfectant.  They describe the various Pseudomonas species found, their resistance profile to commonly used disinfectant and their ability to biofilm formation. The observation that strains able to form biofilms can tolerate disinfection more efficiently is not really surprising. However, that such biofilms may give Listeria monocytogenes a shelter to tolerate better disinfectants should be in terms of food safety of interests. From my prospective studies like presented here are relevant and important to keep hygiene standards and also food safety high and to be able to make recommendations for the industry.

Reviewer 3 Report

Dear Authors,

Manuscript dealing with the considerable tolerance of Pseudomonas spp. in the biofilm and resulting in higher survival of Listeria monotycogenes is a timely and exciting contribution. I positively evaluate the practical direction of the study and the overall interesting experimental approach as well as the achieved results.

However, the text is unfortunately written a little unclearly and chaotically. I also have some other comments on the text (see below) that need to be addressed.

1/ The text must be reviewed in English and also by a native speaker. The text contains ambiguous expressions in many places; commas are missing in sentences according to the rules of English, and the word order is non-standard in some places.

2/ The division of some text into chapters should be revised - in places, the division does not correspond to the content of the paragraphs (the division is unnecessary and somewhat illogical) - see, e.g. chapter 2.2. (but you need to go through the whole text).

3/ Delete L26-34. This is not the text of the manuscript.

4/ L67 - space before the quotation

5/ L68 - percentage values better to one decimal place

6/ L90 "investigate" - small letter "i"

7/ L107 ...as described in PREVIOUS STUDY (30)".

8/ L107 - full stop after the sentence

9/ L112 - delete EFSA in the citation

10/ L117 "in vitro" in italics (also go through other text - e.g., L149, etc.)

11/ L120 - decimal point 0.9

12/ L120 - ...NaCl to OBTAIN a cell density...

13/ L128 - example of a missing comma in a sentence (English).

14/ L134 - redundant expression (see LoD was defined as LoD...); in addition, it is better to write 3xSD

15/ L135 etc. - small letter "mean"

16/ Chapter 2.3. not quite clearly and understandably described - rewrite.

17/ L166 - it is not clear how it was actually done..!? I am to understand that the concentration range of antimicrobials was used for MIC/MBC evaluation after the biofilm test, i.e., the cells "released" from the biofilm were tested??

18/ L272 - duplication - one percent vs. (1%)

19/ Fig3 - "suspension" - do you mean planktonic cells? Probably a better way to express it.

20/ The conclusion does not entirely copy the findings achieved within the study; it must be supplemented and expanded.

21/ The described methodologies must be rewritten in a clear structure. Currently, a large amount of text and complex sentences are used, but the practical implementation remains unclear. (the fundamental problem of the text).

Author Response

Dear Authors,

Manuscript dealing with the considerable tolerance of Pseudomonas spp. in the biofilm and resulting in higher survival of Listeria monotycogenes is a timely and exciting contribution. I positively evaluate the practical direction of the study and the overall interesting experimental approach as well as the achieved results.

However, the text is unfortunately written a little unclearly and chaotically. I also have some other comments on the text (see below) that need to be addressed.

1/ The text must be reviewed in English and also by a native speaker. The text contains ambiguous expressions in many places; commas are missing in sentences according to the rules of English, and the word order is non-standard in some places.

The manuscript has gone though language revision by MDPI’s Standard English language editing service. Changes in the manuscript due to English editing is not marked in the re-submitted manuscript, but the authors have gone through all the suggestions to make sure they improve the language and that they do not change the meaning of the text.

2/ The division of some text into chapters should be revised - in places, the division does not correspond to the content of the paragraphs (the division is unnecessary and somewhat illogical) - see, e.g. chapter 2.2. (but you need to go through the whole text).

            Unnecessary divisions in the text are removed.

3/ Delete L26-34. This is not the text of the manuscript.

            Paragraph deleted.

4/ L67 - space before the quotation

            Space added.

5/ L68 - percentage values better to one decimal place

            One decimal removed from the percentage values.

6/ L90 "investigate" - small letter "i"

            Fixed to small letter.

7/ L107 ...as described in PREVIOUS STUDY (30)".

            Previous study added.

8/ L107 - full stop after the sentence

            Full stop added.

9/ L112 - delete EFSA in the citation

            EFSA is deleted from the reference.

10/ L117 "in vitro" in italics (also go through other text - e.g., L149, etc.)

            In vitro is changed to Italic format in line 118 and 149.

11/ L120 - decimal point 0.9

            Decimal point is fixed.

12/ L120 - ...NaCl to OBTAIN a cell density...

            The word obtain is added.

13/ L128 - example of a missing comma in a sentence (English).

            Comma is added in the sentence: To measure the biofilm formation, the peg lid was transferred to a new 96-well plate containing recovery medium (½ TSB with 1% Tween 20 (0777-1L, VWR Chemicals)) and the biofilm was detached from the pegs by sonication (Branson 5800 Ultrasonic Cleaner) at 40 kHz for 15 min.

14/ L134 - redundant expression (see LoD was defined as LoD...); in addition, it is better to write 3xSD

            Changed to: The Level of Detection (LoD) was defined as meancontol + 3xSDcontrol and was calculated separately for each plate.

15/ L135 etc. - small letter "mean"

            Changed to small letter.

16/ Chapter 2.3. not quite clearly and understandably described - rewrite.

A large part of chapter 2.3 has been rewritten in attempt to clarify the procedure. The authors are aware that this procedure can be complicated to follow. For this reason, we have included the illustration of the procedure in figure S2.

17/ L166 - it is not clear how it was actually done..!? I am to understand that the concentration range of antimicrobials was used for MIC/MBC evaluation after the biofilm test, i.e., the cells "released" from the biofilm were tested?? 

The authors are aware that this procedure can be complicated to follow. Changes has been made in the text in attempt to clarify. For the MIC evaluation it is the challenge plate (the plate containing growth medium with different concentrations of the antimicrobial agents) that is incubated and growth of cells shed during from the biofilm during the challenge that is detected. For the MBC evaluation, 20 µl of the wells in the challenge plate was transferred to a new 96-well plate with new growth medium, and then incubated. So, both MIC and MBC were done on cells that was released from the biofilm during the challenge. No action was applied on the biofilm to actively release the cells at this point. I hope that clarifies this step, but again I will refer to Supplemental Figure S2.

18/ L272 - duplication - one percent vs. (1%)

            Deleted one percent, while 1% is left.

19/ Fig3 - "suspension" - do you mean planktonic cells? Probably a better way to express it.

            The expression “suspension” is exchanged with planktonic growth in Figure 3 and in the main text.

20/ The conclusion does not entirely copy the findings achieved within the study; it must be supplemented and expanded.

The conclusion is expanded to better reflect the findings in the study.

21/ The described methodologies must be rewritten in a clear structure. Currently, a large amount of text and complex sentences are used, but the practical implementation remains unclear. (the fundamental problem of the text).

A large part of the Materials and Methods chapter has been rewritten to clarify how the experiment was done. As some parts of the procedure was rather complex, we included illustrations of this in Supplemental figure S1 and S2. Does the reviewer think it would be better to include these figures in the main manuscript instead of in Supplement? The reason why we placed the figures in Supplemental material is because of the large size. We are however open for advice regarding where to place these figures.

Round 2

Reviewer 1 Report

Thanks authors for the responses. I have no more comments.

Author Response

Thanks to all the reviewers for reviewing our manuscript and providing valuable feedback. The last comment from reviewer #3 is addressed and changes has been made according to the comment.

Reviewer 3 Report

Dear Authors, 

Thank you for all changes within the manuscript that is ready for publication now. I evaluate the presented results as important and interesting for wide readership. 

I have just minor comment:

1/ Chapter 2.3 (L138) - the specification of 13 isolates is missing (Pseudomonas!), and the label "LJP" is missing in case of each strain - it must be in accordance with specification in Table S1!). 

Author Response

(The authors gave the same response as above.)
